# The Interpolated MVU Mechanism For Communication-efficient Private Federated Learning

## Abstract

We consider private federated learning (FL), where a server aggregates differentially private gradient updates from a large number of clients in order to train a machine learning model. The main challenge is balancing privacy with both classification accuracy of the learned model as well as the amount of communication between the clients and server. In this work, we build on a recently proposed method for communication-efficient private FL—the MVU mechanism—by introducing a new interpolation mechanism that can accommodate a more efficient privacy analysis. The result is the new Interpolated MVU mechanism that provides SOTA results on communication-efficient private FL on a variety of datasets.

## 1 Introduction

Machine-learned models leak information about their training data [26]. Private training methods have been developed to train models that provide rigorous guarantees quantifying the amount of information leaked [1, 8, 25]. *Federated learning* (FL) builds on private training to collaboratively train a model among many devices while keeping the data at each device private [20]. To accomplish this, (cross-device) FL requires that devices communicate updates to a server coordinating the training. These updates can be privatized using a differentially private mechanism such as DP-SGD [1] by injecting a controlled amount of noise into the gradient, or update direction, at each step.

To reduce communication overhead in FL, it is also of interest to compress updates before they are transmitted to the server, and lossy compression can also be seen as a way of injecting noise into updates. Most previous work has addressed the challenges of privacy and compression separately, first applying a DP mechanism to privatize the response, and then compressing before transmitting [2, 12].

Recent work [7] introduces the *minimum-variance unbiased* (MVU) mechanism for jointly compressing and ensuring privacy, and experimentally demonstrates that this can lead to better utility-compression trade-offs than other methods which first privatize and then compress. The core of MVU consists of a private mechanism that works for a finite number of scalar inputs. If the input is a bounded continuous scalar, then the solution is to dither to this finite set before applying the core mechanism, and this is further extended to vectors by privacy composition over all coordinates via Rényi DP [21]. Empirically, the MVU mechanism achieves state-of-the-art performance in the local DP setting for both distributed mean estimation and federated learning [7]. However, the analysis in [7] does not benefit from randomization introduced by dithering, and furthermore the extension to vectors leads to suboptimal privacy composition for the $L_2$ geometry, which is often of interest (*e.g.*, working with $L_2$-bounded update vectors such as in DP-SGD).

**Contributions.** Building on a simplified version of the MVU mechanism with only a single scalar input, we propose the *interpolated MVU* (I-MVU) mechanism—a more natural interpolation mechanism to extend MVU to continuous inputs. By its discrete nature, the MVU mechanism can be viewed

as sampling from a particular categorical distribution, and hence can be expressed in exponential family form. The proposed I-MVU mechanism handles continuous inputs by *interpolating* the natural exponential family parameters, rather than directly interpolating the probabilities as in dithering. We introduce a new analysis technique and, by further exploiting special properties of the exponential family, obtain a tight privacy analysis for the vector extension under $L_2$ geometry. Experimentally, we find that under both client-level and user-level DP settings, the I-MVU mechanism provides better privacy-utility trade-off than SignSGD [17] and MVU [7] at an extremely low communication budget of *one bit per gradient dimension*. Moreover, I-MVU achieves close to the same performance as the standard non-compressed Gaussian mechanism [1] for similar levels of $(\epsilon, \delta)$-DP.

## 2  Background and Related Work

**Differential privacy.**  The framework of differential privacy [10] allows rigorous reasoning of privacy leakage through a mechanism $\mathcal{M}$ applied to a dataset $\mathcal{D}$. We say that $\mathcal{M}$ is $(\epsilon, \delta)$-differentially private, denoted $(\epsilon, \delta)$-DP, if for any $\mathcal{D}$, any $\mathbf{x} \in \mathcal{D}$ and any output set $O$, we have:

$$e^{-\epsilon}\mathbb{P}(\mathcal{M}(\mathcal{D} \setminus \mathbf{x}) \in O) - \delta \leq \mathbb{P}(\mathcal{M}(\mathcal{D}) \in O) \leq e^{\epsilon}\mathbb{P}(\mathcal{M}(\mathcal{D} \setminus \mathbf{x}) \in O) + \delta.$$

More generally, the framework of DP seeks to bound the difference in distribution between $\mathcal{M}(\mathcal{D})$ and $\mathcal{M}(\mathcal{D} \setminus \mathbf{x})$ so that a single record $\mathbf{x}$ will not affect the output of the mechanism $\mathcal{M}$ significantly.

A useful variant of DP is Rényi differential privacy (RDP) [21], which instead bounds the Rényi divergence [23] between $\mathcal{M}(\mathcal{D})$ and $\mathcal{M}(\mathcal{D} \setminus \mathbf{x})$ by some $\epsilon$. Formally, we say that $\mathcal{M}$ is $(\alpha, \epsilon)$-RDP if

$$D_\alpha(\mathcal{M}(\mathcal{D}) \,||\, \mathcal{M}(\mathcal{D} \setminus \mathbf{x})) \leq \epsilon \quad \text{and} \quad D_\alpha(\mathcal{M}(\mathcal{D} \setminus \mathbf{x}) \,||\, \mathcal{M}(\mathcal{D})) \leq \epsilon,$$

where $D_\alpha$ denotes the order-$\alpha$ Rényi divergence [21]. Importantly, Rényi DP supports composition of mechanisms in a simple manner: If $\mathcal{M}_1, \ldots, \mathcal{M}_T$ are mechanisms with $\mathcal{M}_t$ being $(\alpha, \epsilon_t)$-RDP for $t = 1, \ldots, T$, then the composition of the $T$ mechanisms is $(\alpha, \sum_{t=1}^{T} \epsilon_t)$-RDP. Another useful property of RDP is its conversion to $(\epsilon, \delta)$-DP [3]: If $\mathcal{M}$ is $(\alpha, \epsilon_\alpha)$-RDP for $\alpha > 1$ then it is also $(\epsilon, \delta)$-DP for any $0 < \delta < 1$ with

$$\epsilon = \epsilon_\alpha + \log\left(\frac{\alpha - 1}{\alpha}\right) - \frac{\log \delta + \log \alpha}{\alpha - 1}. \tag{1}$$

**Federated learning with differential privacy.**  Federated learning (FL) [18, 20] allows distributed training of ML models across multiple clients without centralized data storage. A server coordinates training by acquiring model updates from clients, aggregating them, and then transmitting an updated model back to the clients, with the process repeating until convergence. One promise of FL is data privacy since the updates are computed locally on each client using their own data, and hence no client data is ever explicitly transmitted to the server (or anyone else) throughout the training process. In spite of this, a recent line of work showed that despite the clients never explicitly sharing their data, it is possible to reconstruct training samples from the model updates in a process called *gradient inversion* [11, 31, 32]. This vulnerability remains even if a large number of clients participate in a round using secure aggregation [11, 16, 30].

Differential privacy is a principled method to ensure data privacy in FL as it provides provable guarantees against data reconstruction from the output of a private mechanism [4, 13, 27]. To apply DP to FL training, given a client update $\mathbf{x}$, the client instead sends $\mathcal{M}(\mathbf{x})$ to the server. For a given round, the client's privacy leakage can be computed in terms of *local DP* if the privatized update $\mathcal{M}(\mathbf{x})$ is revealed to the server, or *global DP* if secure aggregation is applied to aggregate the privatized updates before revealing it to the server. The total privacy leakage throughout training can then be computed via RDP composition and conversion to $(\epsilon, \delta)$-DP via Equation 1.

**Communication-efficient private mechanisms.**  Since model updates in FL are high-dimensional vectors of size equal to the number of model parameters, it is also important in practice to compress these updates for communication efficiency. This requirement combined with privacy has led to a series of prior work that designed communication-efficient private mechanisms with application to FL [2, 6, 7, 9, 12, 17, 24, 29]. However, compressing the model update often leads to higher variance and/or biasedness [7, 9], and as a result the model's performance is subpar compared to ones trained using non-compressed DP mechanisms such as the Gaussian mechanism [7, 17]. In this work, we

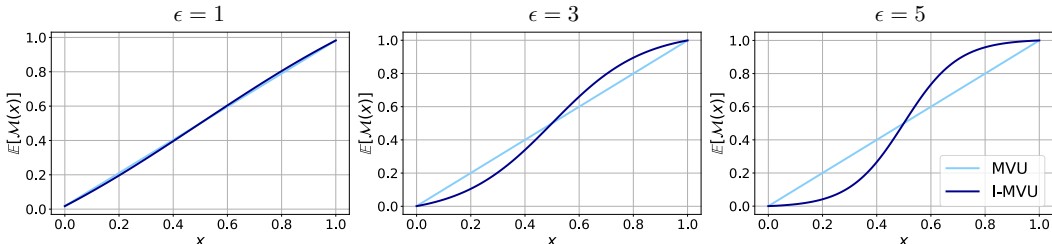

Figure 1: Plot showing the expected value of $\mathcal{M}(x)$ for different values of $x \in [0, 1]$ for the MVU and I-MVU mechanism. While MVU is unbiased in the entire interval $[0, 1]$, I-MVU incurs some bias for $x \neq 0, 1$, especially at higher values of the DP parameter $\epsilon$.

drastically reduce this performance gap and show that replacing the Gaussian mechanism with the proposed interpolated MVU mechanism leads to the same test performance at equal privacy cost when using *one-bit output per coordinate*.

## 3   Interpolated MVU Mechanism

We introduce the *interpolated MVU mechanism*—a communication-efficient differentially-private mechanism with application to private FL training. We also present a novel privacy analysis technique for privatizing vectors with $L_2$ geometry, leading to a drastic improvement in the privacy-utility trade-off over the previously proposed MVU mechanism [7]. To this end, we begin by defining the problem of private-compression and recalling the MVU mechanism.

**Problem description.** Consider the private-compression problem of transmitting a vector $\mathbf{x} \in \mathbb{R}^d$ with bounded $L_2$-norm privately using at most $bd$ bits, with $b$ small enough so that the entire vector $\mathbf{x}$ can be transmitted efficiently. One can reduce this problem to a scalar one by considering how to privately compress $x \in [0, 1]$ using at most $b$ bits, and then scaling the vector $\mathbf{x}$ appropriately to $[0, 1]^d$ and applying the scalar mechanism coordinate-wise.

**The *minimum variance unbiased* (MVU) mechanism [7]** solves the private-compression problem by first discretizing the interval $[0, 1]$ into $B_{\text{in}}$ points $\mathcal{X} = \{x_1 = 0, x_2, \ldots, x_{B_{\text{in}}} = 1\}$ with $x_i := (i - 1)/(B_{\text{in}} - 1)$. If $x = x_i$, the mechanism samples $j \sim \text{Categorical}(\mathbf{p}_i)$ using a probability vector $\mathbf{p}_i \in \Delta^{B_{\text{out}}-1}$ and outputs $\mathcal{M}(x) = a_j \in \mathbb{R}$ where $\{a_1, \ldots, a_{B_{\text{out}}}\}$ is a pre-determined output alphabet. The probability vectors $\mathbf{p}_1, \ldots, \mathbf{p}_{B_{\text{in}}}$ and output alphabet $\{a_1, \ldots, a_{B_{\text{out}}}\}$ are designed so that the mechanism satisfies the following three properties:

1. *$\epsilon$-Differential Privacy*: $e^{-\epsilon}\mathbf{p}_{i',j} \leq \mathbf{p}_{i,j} \leq e^{\epsilon}\mathbf{p}_{i',j}$ for all $i \neq i'$ and all $j$.
2. *Unbiasedness*: $\sum_{j=1}^{B_{\text{out}}} a_j \mathbf{p}_{i,j} = x_i$ for all $i$.
3. *Minimum variance*: $\sum_{i=1}^{B_{\text{in}}} \text{Var}(\mathcal{M}(x_i))$ is minimal among all mechanisms satisfying 1 and 2.

The MVU mechanism can then be applied to all $x \in [0, 1]$ by *randomly dithering* $x$ to the nearest $x_i$ and $x_{i+1}$ such that the dithering is unbiased in expectation. One can also view this dithering procedure as linearly interpolating between $\mathbf{p}_i$ and $\mathbf{p}_{i+1}$. It is straightforward to generalize the mechanism to any bounded $x$ by scaling it to $[0, 1]$ and then applying the MVU mechanism.

For a $d$-dimensional vector $\mathbf{x}$, the MVU mechanism can be applied independently to each coordinate and the privacy cost is $d\epsilon$ by composition if $\mathbf{x} \in [0, 1]^d$ (or in general, if $\|\mathbf{x}\|_\infty$ is bounded). However, the privacy analysis becomes much more complicated for $L_2$-norm bounded vectors—as is often the case for DP-SGD training [1]. We address this problem by expressing the MVU mechanism in exponential family form and interpolating in the natural parameter space, allowing us to use special properties of exponential family distributions to derive tight privacy analysis for the $L_2$ geometry.

**Interpolated MVU mechanism.** As mentioned above, by dithering an input $x \in [x_i, x_{i+1}]$ to $x_i$ or $x_{i+1}$, for general inputs $x \notin \mathcal{X}$, the MVU can be seen as linearly interpolating between the probabilty vectors $\mathbf{p}_i$ and $\mathbf{p}_{i+1}$. Here we improve upon MVU by introducing a better form of interpolation. The

pmf for the categorical distribution with natural parameter $\boldsymbol{\eta}$ can be written as:

$$\mathbb{P}(j|\boldsymbol{\eta}) = \exp(\mathbf{e}_j^\top \boldsymbol{\eta} - A(\boldsymbol{\eta})), \quad A(\boldsymbol{\eta}) = \log\left(\sum_j \exp(\boldsymbol{\eta}_j)\right) \tag{2}$$

where $\mathbf{e}_j$ is the $j$-th standard basis vector. Note that if $\mathbf{p} \in \Delta^{B_{\text{out}}-1}$ then its natural parameter is $\boldsymbol{\eta} = \log \mathbf{p}$. To define the *interpolated MVU* (I-MVU) mechanism, let $\mathbf{p}_1, \mathbf{p}_2 \in \Delta^{B_{\text{out}}-1}$ be sampling probability vectors obtained from the MVU mechanism with $B_{\text{in}} = 2$ and let $\boldsymbol{\eta}_i = \log \mathbf{p}_i$ for $i = 1, 2$ be the natural parameters. Given $x \in [0, 1]$, the I-MVU mechanism samples $j \sim \mathbb{P}(\cdot|\boldsymbol{\eta}(x))$ according to Equation 2 and outputs $a_j$ from the MVU output alphabet, where

$$\boldsymbol{\eta}(x) = (1 - x)\boldsymbol{\eta}_1 + x\boldsymbol{\eta}_2. \tag{3}$$

In other words, instead of linearly interpolating between $\mathbf{p}_1$ and $\mathbf{p}_2$ to construct the sampling probability vector for $x$, we interpolate in the natural parameter space of the categorical distribution. Doing so incurs some bias[1] when $x \notin \{0, 1\}$; see Figure 1 for a plot illustrating this phenomenon. Nevertheless, this bias is small in comparison to the noise induced by differential privacy, and we show empirically that it does not affect the performance of the I-MVU mechanism for FL training.

**Input scaling.** One way to extend I-MVU to arbitrary bounded ranges is to first scale the input to $[0, 1]$ and then apply the mechanism as usual. However, note that the interpolation scheme in Equation 3 is in fact well-defined for any $x \in \mathbb{R}$, and hence the scaled input does not need to be strictly in the range $[0, 1]$. We leverage this property by introducing a scaling factor $\beta$: For $u \in [-C, C]$, the $\beta$-scaled I-MVU mechanism is defined as

$$\mathcal{M}_\beta(u) = \mathcal{M}\left(\frac{1}{2} + \frac{\beta u}{2C}\right),$$

where $\mathcal{M}$ is the plain I-MVU mechanism. Note that this scaling effectively ensures that the input to $\mathcal{M}$ is in the range $[(1 - \beta)/2, (1 + \beta)/2]$, with $\beta = 1$ corresponding to scaling the input to $[0, 1]$. For vectors $\mathbf{u}$ with $\|\mathbf{u}\|_2 \leq C$, the $\beta$-scaled input $\mathbf{x} = \frac{1}{2} + \frac{\beta \mathbf{u}}{2C}$ satisfies $\|\mathbf{x}\|_2 \leq \beta/2$

One advantage for using $\beta$-scaling is that if the distribution of $u$ is highly concentrated near zero, then scaling with $\beta > 1$ ensures that the input to $\mathcal{M}$ is more spread out in the range $[0, 1]$. This ensures that the input distribution more closely reflects the minimum variance requirement (property 3) for the MVU mechanism. For $L_2$-norm bounded vectors $\mathbf{u}$ this is especially true, where the distribution of coordinates of $\mathbf{u}$ is likely concentrated near zero. Consequently, for compressing client updates with bounded $L_2$-norm, $\beta$-scaling with a large $\beta$ is essential for achieving good performance.

### 3.1 Privacy Analysis

We analyze privacy leakage of the I-MVU mechanism for $L_2$-norm bounded vectors in terms of Rényi DP [21]. Our strategy is to first analyze the scalar mechanism and express its Rényi divergence for two differing inputs $x_1$ and $x_2$ as a function of $(x_2 - x_1)^2$ (Lemma 1). Then, by independently applying the mechanism across coordinates, we can sum the Rényi divergence across coordinates and upper bound the total RDP $\epsilon$ as a function of $\|\mathbf{x}_2 - \mathbf{x}_1\|_2^2$ (Theorem 1). Our analysis depends crucially on a measure of information known as *Fisher information*, which we define below for completeness.

**Definition 1.** *Let $f$ be the density function of a distribution parameterized by $x \in \mathbb{R}$. The Fisher information of $x$ contained in a sample $Z \sim f(\cdot|x)$ is:*

$$\mathcal{I}_Z(x) := \mathbb{E}_Z\left[\left(\frac{d}{dx}\log f(Z|x)\right)^2\right]. \tag{4}$$

In our setting, the distribution $\mathbb{P}(\cdot|\boldsymbol{\eta}(x))$ is defined by the private data $x$, and Fisher information measures how much information is revealed about $x$ through a sample $j \sim \mathbb{P}(\cdot|\boldsymbol{\eta}(x))$. It is noteworthy that such a reasoning has also been used to define Fisher information as a privacy metric [14].

**Lemma 1.** *Let $M = \sup_{x \in \mathbb{R}} \mathcal{I}_Z(x)$ be an upper bound on the Fisher information of the mechanism $\mathcal{M}$. Then for any $x_1, x_2 \in \mathbb{R}$:*

$$D_\alpha(\mathbb{P}(\cdot|\boldsymbol{\eta}(x_1)) \,||\, \mathbb{P}(\cdot|\boldsymbol{\eta}(x_2))) \leq \alpha M(x_2 - x_1)^2/2. \tag{5}$$

---

[1]In spite of this, we still name our mechanism I-MVU for its connection to the MVU mechanism.

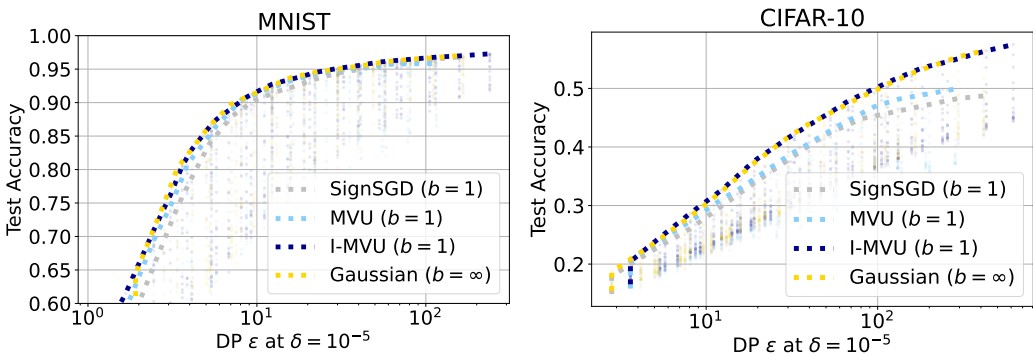

Figure 2: Privacy vs. accuracy plot for the client-level DP scenario on MNIST (left) and CIFAR-10 (right). Each point represents a single hyperparameter setting and the Pareto frontier is shown in dashed line. Across the entire range of $\epsilon$, I-MVU consistently performs as well as the non-compressed Gaussian mechanism while requiring only one bit communication per update coordinate.

**Theorem 1.** *Let $M$ be the Fisher information constant from Lemma 1. Suppose that $\mathbf{x}_1, \mathbf{x}_2 \in \mathbb{R}^d$ satisfy $\|\mathbf{x}_2 - \mathbf{x}_1\|_2 \leq C$. Then the I-MVU mechanism is $(\alpha, \alpha M C^2/2)$-RDP for all $\alpha > 1$.*

**Proof sketch.** We first derive the Taylor series expression for the Rényi divergence between $\mathbb{P}(\cdot|\boldsymbol{\eta}(x_1))$ and $\mathbb{P}(\cdot|\boldsymbol{\eta}(x_2))$. Since Rényi divergence is minimized and is equal to 0 when $x_1 = x_2$, the zeroth-order and first-order terms in the Taylor series are 0. The coefficient for the second-order term is given by the Fisher information $\mathcal{I}_Z(x_1)$ [15], and thus we give a numerical method to compute $M = \sup_{x \in \mathbb{R}} \mathcal{I}_Z(x)$ in Appendix B and use it in Equation 5 to bound the RDP $\epsilon$. Full proofs of Lemma 1 and Theorem 1 are provided in Appendix A.

# 4 Experiments

We evaluate the I-MVU mechanism for federated learning under the local DP setting, *i.e.*, clients transmit the privately compressed model update $\mathcal{M}(\mathbf{x})$ to the server *before* aggregation. We consider private mechanisms that output *one bit per coordinate* of the update vector. This extreme level of compression reflects realistic constraints in FL and is very challenging for existing mechanisms. Previous work [7] found that the two most competitive baselines are the MVU mechanism with $b = 1$ bit communication budget and SignSGD [17]. The latter applies the Gaussian mechanism for gradient perturbation [1] and then takes the sign of the perturbed gradient to obtain one-bit per coordinate.

## 4.1 Client-level DP

We first evaluate under the *client-level DP* setting on MNIST and CIFAR-10 [19]. Here, the privacy analysis guarantees that the learning algorithm is differentially private with respect to the removal of any client. We divide the training set among the clients with client sample size 1. Each client performs a single local gradient update in every FL round. This setting is equivalent to DP-SGD training [1] but with the Gaussian mechanism replaced by a communication-efficient private mechanism.

**Training details.** Following [7], we train a linear model on top of ScatterNet features [28], which remains to date the state-of-the-art DP model for MNIST and CIFAR-10 without leveraging any public data. We perform a grid search over hyperparameters such as number of update rounds, step size, gradient norm clip, and mechanism parameters $\sigma$ (for Gaussian and SignSGD) and $\epsilon$ (for MVU and I-MVU). We use the same hyperparameter values reported in Tables 3 and 4 in [7].

**Result.** Figure 2 shows the privacy vs. test accuracy curve on MNIST (left) and CIFAR-10 (right). Privacy is measured in terms of $(\epsilon, \delta)$-DP at $\delta = 10^{-5}$. Each point in the scatter plot corresponds to a single hyperparameter setting and the dashed line shows the Pareto frontier of optimal privacy-accuracy trade-off. The yellow line corresponds to the standard Gaussian mechanism without compression, which attains the best test accuracy at any given privacy budget $\epsilon$. Both SignSGD and MVU are competitive, achieving close to the same level of accuracy as the Gaussian mechanism, but a non-negligible gap remains, especially on CIFAR-10. In contrast, I-MVU attains nearly the same performance as the Gaussian mechanism at all values of $\epsilon$ on both MNIST and CIFAR-10.

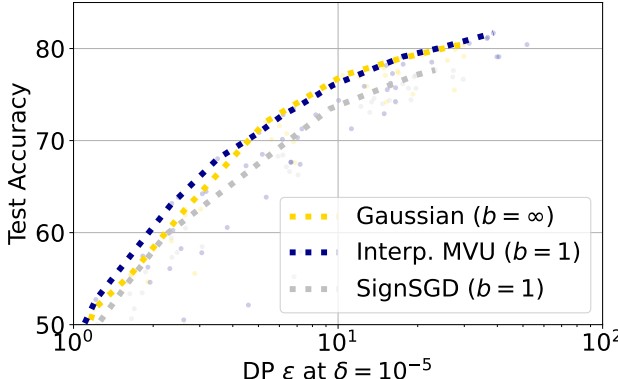

Figure 3: Privacy vs. accuracy plot for the sample-level DP scenario on FEMNIST. Each point represents a single hyperparameter setting and the Pareto frontier is shown in dashed line. I-MVU with one-bit communication budget per coordinate consistently performs better than SignSGD and is competitive with the non-compressed Gaussian baseline across the entire range of $\epsilon$.

Since MVU and I-MVU are near-identical mechanisms, we argue that the performance gain comes primarily from the tight privacy analysis for $L_2$ geometry using Fisher information (Section 3.1).

## 4.2 Sample-level DP

Next, we evaluate under the *sample-level DP* setting on the FEMNIST dataset [5] for classifying written characters into 62 distinct classes. Privacy analysis guarantees that the learning algorithm is differentially private with respect to the removal of *any training sample from a client*. The dataset has a pre-defined train split with $3,500$ clients, from which we randomly select $3,150$ clients for training and the remaining 350 clients for testing. A set of 5 clients is selected in each training round, who then performs full batch gradient descent for a single local gradient update to compute the update vector. The update vector is privatized using a communication-efficient private mechanism and transmitted to the server.

**Training details.** We train a simple 4-layer convolutional network for classification. The model achieves $84\%$ accuracy when trained non-privately. The client optimizer is SGD with a learning rate of $0.1$ and no momentum. The server implements FedAvg [20] with a momentum of $0.9$. We perform a grid search on the local and server learning rates, the clipping factor, the noise multiplier $\sigma$ for both Gaussian and SignSGD baselines, and the $\epsilon$ and scale hyperparameters for I-MVU. The hyperparameter ranges are given in Tables 1, 2 and 3 in the appendix. In particular, SignSGD requires much lower server-side learning rates since the updates (in $\{\pm 1\}$) have higher magnitude.

**Result.** We show the privacy-accuracy trade-off for FEMNIST in Figure 3. Each point in the scatter plot represents a single hyperparameter setting and the Pareto frontier (dashed line) represents the optimal privacy-accuracy trade-off. The DP privacy budget $\epsilon$ is given at $\delta = 10^{-5}$. We observe that I-MVU (blue dashed line) performs better than SignSGD (silver dashed line) for the same communication budget of one bit per update coordinate across the entire range of considered privacy budgets $\epsilon$. Moreover, I-MVU performs on par with the non-compressed Gaussian baseline (yellow dashed line), where clients perform local DP-SGD without compressing model updates.

## 5 Conclusion

We proposed the Interpolated MVU (I-MVU) mechanism that drastically reduces the amount of uplink communication in (cross-device) FL while providing differential privacy guarantees. Our proposal builds on the recently introduced MVU mechanism to extend it to continuous-valued vectors with $L_2$ geometry using a more efficient privacy analysis. Under both client-level and sample-level local DP settings, I-MVU with an extreme compression level of one bit per update coordinate attains close to the performance of the non-compressed Gaussian mechanism. Given this strong empirical performance, we advocate for I-MVU as a practical tool for communication-efficient private FL.

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

## A Proofs

**Lemma 1.** *Let $M = \sup_{x \in \mathbb{R}} \mathcal{I}_Z(x)$ be an upper bound on the Fisher information of the mechanism $\mathcal{M}$. Then for any $x_1, x_2 \in \mathbb{R}$:*

$$D_\alpha(\mathbb{P}(\cdot | \boldsymbol{\eta}(x_1)) \ || \ \mathbb{P}(\cdot | \boldsymbol{\eta}(x_2))) \leq \alpha M (x_2 - x_1)^2 / 2.$$

*Proof.* We first derive an explicit form for the Fisher information. Let $f(\mathbf{z}; x)$ denote the pmf in Equation 2 for any $\mathbf{z} \in \{\mathbf{e}_1, \ldots, \mathbf{e}_{B_{\text{out}}}\}$. The log pmf is:

$$\log f(\mathbf{z}; x) = \mathbf{z}^\top \boldsymbol{\eta}(x) - A(\boldsymbol{\eta}(x)) \tag{6}$$

Taking derivative with respect to $x$ gives:

$$\frac{d}{dx} \log f(\mathbf{z}; x) = (\mathbf{z} - \sigma(\boldsymbol{\eta}(x)))^\top (\boldsymbol{\eta}_2 - \boldsymbol{\eta}_1),$$

$$\left( \frac{d}{dx} \log f(\mathbf{z}; x) \right)^2 = (\boldsymbol{\eta}_2 - \boldsymbol{\eta}_1)^\top (\mathbf{z} - \sigma(\boldsymbol{\eta}(x)))(\mathbf{z} - \sigma(\boldsymbol{\eta}(x)))^\top (\boldsymbol{\eta}_2 - \boldsymbol{\eta}_1),$$

where $\sigma$ denotes the sigmoid function. Taking expectation over $\mathbf{z}$ gives the Fisher information:

$$\mathcal{I}_Z(x) = (\boldsymbol{\eta}_2 - \boldsymbol{\eta}_1)^\top U (\boldsymbol{\eta}_2 - \boldsymbol{\eta}_1), \tag{7}$$

with $U = \text{diag}(\sigma(\boldsymbol{\eta}(x))) - \sigma(\boldsymbol{\eta}(x))\sigma(\boldsymbol{\eta}(x))^\top$.

To derive the upper bound, we first define a function $F$ for the Rényi divergence of the mechanism for a fixed $x_1$ and varying $x_2$:

$$F_\alpha(x_2; x_1) = D_\alpha(\mathbb{P}(\cdot | \boldsymbol{\eta}(x_1)) \ || \ \mathbb{P}(\cdot | \boldsymbol{\eta}(x_2))). \tag{8}$$

By Taylor's theorem, we can express $F$ as:

$$F_\alpha(x_2; x_1) = F_\alpha(x_1; x_1) + (x_2 - x_1)F_\alpha'(x_1; x_1) + (x_2 - x_1)^2 F_\alpha''(\xi; x_1)/2$$

for some $\xi \in [x_1, x_2]$. Note that $F_\alpha(x_1; x_1) = 0$ and $F_\alpha'(x_1; x_1) = 0$ (since $x_1$ is the global minimizer of $F_\alpha(\cdot; x_1)$), so $F$ is locally a quadratic function:

$$F_\alpha(x_2; x_1) = (x_2 - x_1)^2 F_\alpha''(\xi; x_1)/2. \tag{9}$$

Since $f$ is the pmf of an exponential family distribution, we can use the closed form expression [22] for Rényi divergence of exponential family distributions to express $F$ and its derivatives:

$$F_\alpha(\xi; x_1) = \frac{1}{\alpha - 1}[A(\alpha\boldsymbol{\eta}(x_1) + (1 - \alpha)\boldsymbol{\eta}(\xi)) - \alpha A(\boldsymbol{\eta}(x_1)) - (1 - \alpha)A(\boldsymbol{\eta}(\xi))]$$

$$F_\alpha'(\xi; x_1) = (\sigma(\boldsymbol{\eta}(\xi)) - \sigma(\alpha\boldsymbol{\eta}(x_1) + (1 - \alpha)\boldsymbol{\eta}(\xi)))^\top (\boldsymbol{\eta}_2 - \boldsymbol{\eta}_1)$$

$$F_\alpha''(\xi; x_1) = (\boldsymbol{\eta}_2 - \boldsymbol{\eta}_1)^\top (V + (\alpha - 1)W)(\boldsymbol{\eta}_2 - \boldsymbol{\eta}_1)$$

where $V = \text{diag}(\sigma(\boldsymbol{\eta}(\xi))) - \sigma(\boldsymbol{\eta}(\xi))\sigma(\boldsymbol{\eta}(\xi))^\top$, $W = \text{diag}(\sigma(\boldsymbol{\eta}(x'))) - \sigma(\boldsymbol{\eta}(x'))\sigma(\boldsymbol{\eta}(x'))^\top$, and $x' = \alpha x_1 + (1 - \alpha)\xi$. Hence $F_\alpha''(\xi; x_1) = \mathcal{I}_Z(\xi) + (\alpha - 1)\mathcal{I}_Z(x')$ by Equation 7. Upper bounding $\mathcal{I}_Z(\xi)$ and $\mathcal{I}_Z(x')$ by $M := \sup_{x \in \mathbb{R}} \mathcal{I}_Z(x)$ and combining with Equation 9 gives the desired result. $\square$

**Theorem 1.** *Let $M$ be the Fisher information constant from Lemma 1. Suppose that $\mathbf{x}_1, \mathbf{x}_2 \in \mathbb{R}^d$ satisfy $\|\mathbf{x}_2 - \mathbf{x}_1\|_2 \leq C$, then the I-MVU mechanism is $(\alpha, \alpha M C^2 / 2)$-RDP for all $\alpha > 1$.*

*Proof.* Let $\mathbf{a} = \mathcal{M}(\mathbf{x}) \in \{a_1, \ldots, a_{B_{\text{out}}}\}^d$ be the output of the vector I-MVU mechanism that independently applies the scalar mechanism to each coordinate. Then:

$$
D_\alpha(\mathcal{M}(\mathbf{x}_1) \,\|\, \mathcal{M}(\mathbf{x}_2)) = \frac{1}{\alpha - 1} \log \sum_{\mathbf{a} \in \{a_1, \ldots, a_{B_{\text{out}}}\}^d} \prod_{k=1}^d \frac{\mathbb{P}(\mathbf{a}_k | \boldsymbol{\eta}((\mathbf{x}_2)_k))^\alpha}{\mathbb{P}(\mathbf{a}_k | \boldsymbol{\eta}((\mathbf{x}_1)_k))^{\alpha-1}}
$$

$$
= \sum_{k=1}^d \frac{1}{\alpha - 1} \log \sum_{\mathbf{a}_k \in \{a_1, \ldots, a_{B_{\text{out}}}\}} \frac{\mathbb{P}(\mathbf{a}_k | \boldsymbol{\eta}((\mathbf{x}_2)_k))^\alpha}{\mathbb{P}(\mathbf{a}_k | \boldsymbol{\eta}((\mathbf{x}_1)_k))^{\alpha-1}} \quad \text{by independence}
$$

$$
= \sum_{k=1}^d D_\alpha(\mathbb{P}(\cdot | \boldsymbol{\eta}((\mathbf{x}_1)_k)) \,\|\, \mathbb{P}(\cdot | \boldsymbol{\eta}((\mathbf{x}_2)_k)))
$$

$$
\leq \sum_{k=1}^d \alpha M((\mathbf{x}_2)_k - (\mathbf{x}_1)_k)^2 / 2 \quad \text{by Lemma 1}
$$

$$
= \alpha M \|\mathbf{x}_2 - \mathbf{x}_1\|_2^2 / 2.
$$

$\square$

# B Computing Fisher Information

In this section we describe a method for computing $M = \sup_{x \in \mathbb{R}} \mathcal{I}_Z(x)$. We first define a condition for $\boldsymbol{\eta}_1, \boldsymbol{\eta}_2$ that allows us to reduce this problem to maximizing $\mathcal{I}_Z(x)$ over a bounded range $[x_{\min}, x_{\max}]$.

**Definition 2.** *Two vectors $\boldsymbol{\eta}_1, \boldsymbol{\eta}_2 \in \mathbb{R}^B$ are said to be* anadromic *if for all $j = 1, \ldots, B$, we have $(\boldsymbol{\eta}_1)_j = (\boldsymbol{\eta}_2)_{B-j+1}$.*

The following technical lemma proves several useful properties that hold when $\boldsymbol{\eta}_1$ and $\boldsymbol{\eta}_2$ are anadromic.

**Lemma 2.** *Suppose that $\boldsymbol{\eta}_1, \boldsymbol{\eta}_2 \in \mathbb{R}^B$ are anadromic. Let $\boldsymbol{\theta} = \boldsymbol{\eta}_2 - \boldsymbol{\eta}_1$ and suppose that $j^+ = \arg\max_j \boldsymbol{\theta}_j, j^- = \arg\min_j \boldsymbol{\theta}_j$ are unique. Then the following hold:*

*(i) $\boldsymbol{\theta}_j = -\boldsymbol{\theta}_{B-j+1}$ for all $j$, and hence $j^- = B - j^+ + 1$.*
*(ii) $\boldsymbol{\eta}(x)_j = \boldsymbol{\eta}(1 - x)_{B-j+1}$ for all $j$.*
*(iii) $\sigma(\boldsymbol{\eta}(x)) \to \mathbf{e}_{j^+}$ as $x \to \infty$ and $\sigma(\boldsymbol{\eta}(x)) \to \mathbf{e}_{j^-}$ as $x \to -\infty$.*
*(iv) $\mathcal{I}_Z(x) = \mathcal{I}_Z(1 - x)$ for all $x \in \mathbb{R}$.*
*(v) $x = 1/2$ is a stationary point for $\mathcal{I}_Z(x)$.*
*(vi) If $\sigma(\boldsymbol{\eta}(x))_{j^+} \geq 1/2$ then $\mathcal{I}_Z(x) \leq 4\boldsymbol{\theta}_{j^+}^2 \sigma(\boldsymbol{\eta}(x))_{j^+}(1 - \sigma(\boldsymbol{\eta}(x))_{j^+})$.*

*Proof.* (i) Since $\boldsymbol{\eta}_1$ and $\boldsymbol{\eta}_2$ are anadromic,

$$
\boldsymbol{\theta}_j = (\boldsymbol{\eta}_2)_j - (\boldsymbol{\eta}_1)_j = (\boldsymbol{\eta}_2)_j - (\boldsymbol{\eta}_2)_{B-j+1} = -((\boldsymbol{\eta}_2)_{B-j+1} - (\boldsymbol{\eta}_1)_j) = -\boldsymbol{\theta}_{B-j+1}.
$$

In particular, $\arg\max_j \boldsymbol{\theta}_j = B - (\arg\min_j \boldsymbol{\theta}_j) + 1$.

(ii) $\boldsymbol{\eta}(x)_j = (1 - x)(\boldsymbol{\eta}_1)_j + x(\boldsymbol{\eta}_2)_j = (1 - x)(\boldsymbol{\eta}_1)_{B-j+1} + x(\boldsymbol{\eta}_2)_{B-j+1} = \boldsymbol{\eta}(1 - x)_{B-j+1}$.

(iii) Let $\bar{\boldsymbol{\eta}} = (\boldsymbol{\eta}_1 + \boldsymbol{\eta}_2)/2$ so that $\boldsymbol{\eta}(x) = \bar{\boldsymbol{\eta}} + (x - 1/2)\boldsymbol{\theta}$ for all $x \in \mathbb{R}$. It is clear that $\sigma(\boldsymbol{\eta}(x)) \to \mathbf{e}_{j^+}$ as $x \to \infty$ since $j^+$ is unique. A similar argument shows that $\sigma(\boldsymbol{\eta}(x)) \to \mathbf{e}_{j^-}$ as $x \to -\infty$.

(iv) Using the expression of $\mathcal{I}_Z(x)$ in the proof of Lemma 1, we get

$$
\mathcal{I}_Z(x) = \sum_j \boldsymbol{\theta}_j^2 \sigma(\boldsymbol{\eta}(x))_j - \left( \sum_j \boldsymbol{\theta}_j \sigma(\boldsymbol{\eta}(x))_j \right)^2
$$

$$
= \sum_j \boldsymbol{\theta}_{B-j+1}^2 \sigma(\boldsymbol{\eta}(1 - x))_{B-j+1} - \left( \sum_j \boldsymbol{\theta}_{B-j+1} \sigma(\boldsymbol{\eta}(1 - x))_{B-j+1} \right)^2 \quad \text{by (i) and (ii)}
$$

$$
= \mathcal{I}_Z(1 - x).
$$

(v) Differentiating $\mathcal{I}_Z(x)$ and using the above argument gives:

$$
\begin{aligned}
\mathcal{I}_Z'(x) &= (\boldsymbol{\theta}^3)^\top \sigma(\boldsymbol{\eta}(x)) - 3\boldsymbol{\theta}^\top \sigma(\boldsymbol{\eta}(x))(\boldsymbol{\theta}^2)^\top \sigma(\boldsymbol{\eta}(x)) + 2(\boldsymbol{\theta}^\top \sigma(\boldsymbol{\eta}(x)))^3 \\
&= -(\boldsymbol{\theta}^3)^\top \sigma(\boldsymbol{\eta}(1-x)) + 3\boldsymbol{\theta}^\top \sigma(\boldsymbol{\eta}(1-x))(\boldsymbol{\theta}^2)^\top \sigma(\boldsymbol{\eta}(1-x)) - 2(\boldsymbol{\theta}^\top \sigma(\boldsymbol{\eta}(1-x)))^3 \\
&= -\mathcal{I}_Z'(1-x).
\end{aligned}
$$

Then $\mathcal{I}_Z'(1/2) = -\mathcal{I}_Z'(1/2)$, so $\mathcal{I}_Z'(1/2) = 0$ and $x = 1/2$ is a stationary point.

(vi) Using the fact that $0 \le \boldsymbol{\theta}_j^2 \le \boldsymbol{\theta}_{j+}^2$ for all $j$ and

$$
\sum_j \boldsymbol{\theta}_j \sigma(\boldsymbol{\eta}(x))_j \ge \boldsymbol{\theta}_{j+} \sigma(\boldsymbol{\eta}(x))_{j+} + \boldsymbol{\theta}_{j-}(1-\sigma(\boldsymbol{\eta}(x))_{j+}) = \boldsymbol{\theta}_{j+}\sigma(\boldsymbol{\eta}(x))_{j+} - \boldsymbol{\theta}_{j+}(1-\sigma(\boldsymbol{\eta}(x))_{j+}) \ge 0,
$$

we have:

$$
\begin{aligned}
\mathcal{I}_Z(x) &= \sum_j \boldsymbol{\theta}_j^2 \sigma(\boldsymbol{\eta}(x))_j - \left(\sum_j \boldsymbol{\theta}_j \sigma(\boldsymbol{\eta}(x))_j\right)^2 \\
&\le \boldsymbol{\theta}_{j+}^2 - (\boldsymbol{\theta}_{j+}\sigma(\boldsymbol{\eta}(x))_{j+} - \boldsymbol{\theta}_{j+}(1-\sigma(\boldsymbol{\eta}(x))_{j+}))^2 \\
&= \boldsymbol{\theta}_{j+}^2(1 - (2\sigma(\boldsymbol{\eta}(x))_{j+} - 1)^2) \\
&= 4\boldsymbol{\theta}_{j+}^2 \sigma(\boldsymbol{\eta}(x))_{j+}(1 - \sigma(\boldsymbol{\eta}(x))_{j+}).
\end{aligned}
$$

$\square$

**Algorithm.** To use Lemma 2 to compute $M$, we first compute $I^* = \mathcal{I}_Z(1/2)$ since $x = 1/2$ is a stationary point by Lemma 2(v). By setting

$$
4\boldsymbol{\theta}_{j+}^2 \sigma(\boldsymbol{\eta}(x))_{j+}(1 - \sigma(\boldsymbol{\eta}(x))_{j+}) \le I^*
$$

and solving this quadratic equation for $\sigma(\boldsymbol{\eta}(x))_{j+}$, we can use the bound in Lemma 2(vi) to obtain that $\mathcal{I}_Z(x) \le I^*$ when $\sigma(\boldsymbol{\eta}(x))_{j+} \ge \left(1 + \sqrt{1 - I^*/\boldsymbol{\theta}_{j+}^2}\right)/2 \ge 1/2$. Since $\sigma(\boldsymbol{\eta}(x))_{j+} \to 1$ as $x \to \infty$ by (iii), we can determine the value $x_{\max}$ for which $\mathcal{I}_Z(x) \le I^*$ when $x \ge x_{\max}$. By (iv), $x_{\min} = 1 - x_{\max}$ satisfies $\mathcal{I}_Z(x) \le I^*$ when $x \le x_{\min}$. We can then do line search in $[x_{\min}, x_{\max}]$ (or equivalently, in $[1/2, x_{\max}]$ by Lemma 2(iv)) to obtain $M$.

## C   Hyperparameters for FEMNIST

| Hyperparameter | Values |
| --- | --- |
| $\varepsilon$ | 0.25, 0.5, 0.75, 1, 2, 3, 5, 6, 7, 8, 9, 10 |
| Server-side learning rate | 0.5, 1, 2 |
| Scaling factor $\beta$ | 32, 64, 128 |

Table 1: Hyperparameter range for I-MVU on FEMNIST.

| Hyperparameter | Values |
| --- | --- |
| $\sigma$ | 0.6, 0.8, 1, 2, 4, 6, 8, 10, 16, 32, 64, 128 |
| Server-side learning rate | 0.5, 1, 2 |
| Clipping factor | 0.5, 1, 2 |

Table 2: Hyperparameter range for Gaussian on FEMNIST.

| Hyperparameter | Values |
| --- | --- |
| $\sigma$ | 0.6, 0.8, 1, 2, 4, 6, 8, 10, 16, 32, 64, 128 |
| Server-side learning rate | 0.0001, 0.001, 0.01 |
| Clipping factor | 0.5, 1, 2 |

Table 3: Hyperparameter range for SignSGD on FEMNIST.

