# OpenReview forum: "The Interpolated MVU Mechanism For Communication-efficient Private Federated Learning"
_NeurIPS.cc/2022/Workshop/Federated_Learning — FL-NeurIPS 2022 Poster_

### Official Review · Reviewer_tmZK · 2022-10-18

This paper proposed the I-MVU mechanism that reduces communication cost while providing a differential privacy guarantee. This paper extends the previously introduced MVU mechanism to continuous-valued vectors. This paper could achieve a good tradeoff between utility and privacy. This paper conducted experiments to support their claims. This paper is well written. I don't have any other concerns.

---

### Official Review · Reviewer_q9e8 · 2022-10-18
**Developed differential privacy could be interesting.**

I am not an expert on the privacy aspect of Federated Learning specifically. However, I think developed can be applied on top of many different Federated Learning algorithms and due to simple analysis, we can derive privacy bounds for those algorithms.

 One question that I have and would be great if authors can clarify is whether can this newly developed mechanism also provide guarantees for pairwise privacy between users.

---

### Decision · Program_Chairs · 2022-10-20

Accept (Poster)